# VideoClusterNet: Self-Supervised and Adaptive Face Clustering for Videos

## Abstract

With the rise of digital media content production, the need for analyzing movies and TV series episodes to locate the main cast of characters precisely is gaining importance. Specifically, Video Face Clustering aims at grouping together detected video face tracks with common facial identities. This problem is very challenging due to the large range of pose, expression, appearance, and lighting variations of a given face across video frames. Generic pre-trained Face Identification (ID) models fail to adapt well to the video production domain, given its high dynamic range content and also unique cinematic style. Furthermore, traditional clustering algorithms depend on hyperparameters requiring individual tuning across datasets. In this paper, we present a novel video face clustering approach that learns to adapt a generic face ID model to new video face tracks in a fully self-supervised fashion. We also propose a parameter-free clustering algorithm that is capable of automatically adapting to the finetuned model's embedding space for any input video. Due to the lack of comprehensive movie face clustering benchmarks, we also present a first-of-kind movie dataset: MovieFaceCluster. Our dataset is handpicked by film industry professionals and contains extremely challenging face ID scenarios. Experiments demonstrate our method's effectiveness in handling difficult mainstream movie scenes on our benchmark dataset and state-of-the-art performance on traditional TV series datasets.

## 1 Introduction

Video Face Clustering can be defined as the task of grouping together human faces in a video among common identities. It contributes significantly to several other research domains, such as video scene captioning Rohrbach et al. (2017), video question answering Tapaswi et al. (2016), and video understanding Vicol et al. (2018). Having an understanding of the spatial location, face size, and identity of the characters that appear in specific scenes is essential for all the aforementioned tasks. Clustering faces in a video is a challenging unsupervised problem that has garnered a lot of interest over the past few decades Satoh et al. (1999); Pham et al. (2009); Zhou et al. (2015); Wu et al. (2013b). Given the rise in the creation of video production content and the subsequent need for its analysis, face clustering in the movie/TV series domain has garnered significant interest in the last couple of years Sharma et al. (2019); Tapaswi et al. (2019). It serves as an effective editing tool for movie post-production personnel, helping them select scenes with a specific group of characters, among other benefits. We thus primarily focus on the video production content domain for evaluating our proposed method, given its closeness to real-world scenarios and use cases.

The video production content domain often provides an unique set of challenges for face clustering, in terms of large variations in facial pose, lighting, expression and appearance w.r.t. a given character across its entire duration (Fig. 1). In specific domains with high-quality standards, such as movies that possess an unique cinematic style[1], performance of face identification (ID) models trained on generic large-scale datasets is often limited for such domains (Tab. 5). Furthermore, hand labeling a large cast of characters, often present in movies/TV series, can be very time-consuming and error-prone. As a result, the inherent challenges in video face clustering and difficulties in hand labeling often limit video-specific model training. In this paper, we propose an algorithm that successfully

---

[1]Particular movie features are high resolution, high dynamic range, and large facial attribute variations.

tackles these limitations. Specifically, our proposed method adapts a generic face ID model to a specific set of faces and their observed variations in a given video in a fully self-supervised fashion.

Also, traditional deep face clustering algorithms Zhou et al. (2022c) present certain limitations, which can be categorized into two main types. The first group Defays (1977); Ester et al. (1996) adopts a bottom-up approach to clustering and incorporates a pre-defined distance function to compare face embeddings, thus requiring an user-defined threshold to specify a positive match. The second group Lloyd (1982); Comaniciu & Meer (2002) follows a top-down approach and requires the number of known clusters or minimum cluster sample count as input for its initialization. Both of these groups thus have shortcomings, in terms of requiring non-intuitive user defined parameters. As an essential component of our idea, we present a novel clustering algorithm that improves on these limitations. It is fully automated without the need for any user input and uses a distance metric that is optimized for the given model-learned embedding space.

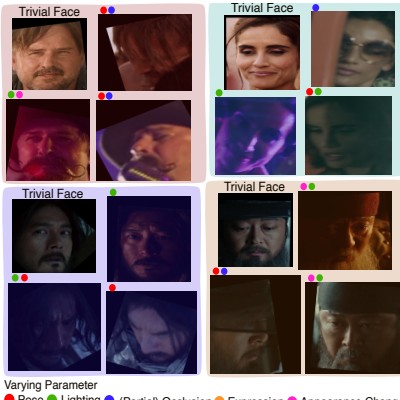

Figure 1: Select hard case clusters predicted using our algorithm from within **MovieFaceCluster** dataset. Trivial face represents an easy ID sample for each cluster.

Overall, our proposed method VideoClusterNet can be divided into two main stages. The first stage involves a fully self-supervised learning (SSL) based finetuning of a generic face ID model on a given set of video faces. Finetuning is formulated as an iterative optimization task facilitated through alternating stages of model finetuning and coarse face track matching. The SSL finetuning bootstraps itself by soft grouping together high-confidence matching tracks at regular training intervals. The second stage involves a track clustering algorithm that adopts the loss function used for model SSL finetuning as a distance metric. Our clustering algorithm computes a custom matching threshold for each track and combines tracks with high-confidence matches in an iterative bottom-up style.

Given that the video content production domain provides unique in-the-wild challenges regarding video face clustering, the academic research community lacks a standardized video dataset benchmark for real-world performance evaluation (Appendix E). Thus, we also present a novel video face clustering dataset, which incorporates challenging movies hand-selected by experienced film post-production specialists. We conduct extensive experiments of our proposed method on this dataset to validate its effectiveness for character clustering in mainstream movies. In addition, we provide results on selects benchmark datasets, showing that our method attains state-of-the-art performance.

In summary, **we propose the following contributions:** 1) A fully self-supervised video face clustering algorithm, which progressively learns robust identity embeddings for all faces within a given face video dataset, facilitated via iterative soft matching of faces across pose, illumination, and expression variations observed in the dataset. 2) A self-supervised model finetuning approach that unlike prior works relies only on positive match pairs, removing any dependence on manual ground truth labels or use of temporal track constraints to obtain negative match pairs. 3) A deep learning-based similarity metric for face clustering, which automatically adapts to a given model's learned embedding space. 4) A novel parameter-free video face clustering method that requires neither user-defined thresholds nor an initial number of clusters. 5) A new comprehensive movie face clustering benchmark dataset to better evaluate video face clustering algorithms on real-world challenges.

## 2 RELATED WORK

We review prior work in video-based face clustering and list out some deep learning metric and self-supervised learning based methods since they form an important component in our approach.

**Auxiliary labels assisted Video Face Clustering:** Single frame-based face clustering has drawn a lot of attention in the past few decades. For a detailed survey, please refer to Zhou et al. (2022c). For the video domain, early work focused on using additional information available from TV series episodes/movies. Specifically, methods such as Satoh et al. (1999); Berg et al. (2004); Cour et al. (2010); Everingham et al. (2009); Tang et al. (2015); Ozkan & Duygulu (2006); Pham et al. (2009); Everingham et al. (2006) utilize aligned captions, transcripts, dialogues, or a combination of the above with detected faces to perform identity clustering.

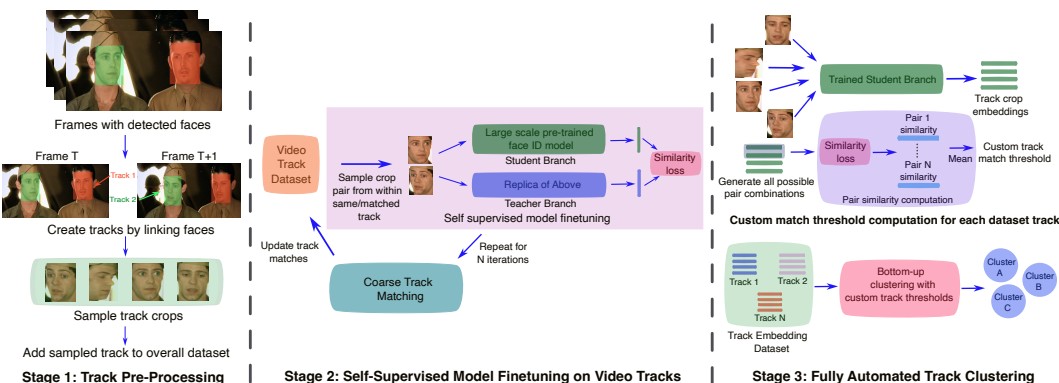

Figure 2: **Overview of VideoClusterNet:** Given the temporal continuity in the video domain, faces detected in consecutive frames are first locally grouped into tracks using a motion tracking algorithm. A large-scale pre-trained face ID model is finetuned on these tracks using temporal self-supervision (w/ only positive pairing), via learning through natural face variations available within each track. Then, the finetuning is bootstrapped by soft-matching tracks across common identities. Performing these two steps alternatively helps the model better understand the given set of faces. A final clustering algorithm based on a model-learned similarity metric groups common identity tracks.

**Contextual Information based Video Face Clustering:** Following work using supplementary labels, methods such as Zhang et al. (2013); El Khoury et al. (2010) leverage contextual information, e.g., clothing and surrounding scene contents, while Paul et al. (2014) use aligned audio to help localize faces. Alternatively, Zhou et al. (2015) incorporate gender information along with temporal constraints through face motion tracking. Unlike previous approaches, our method requires no explicit contextual information.

**Video Face Clustering using temporal feature aggregation, 3D convolutions:** Another line of work, such as in Liu et al. (2019); Gong et al. (2019), incorporates mechanisms to aggregate deep learning-based features of a given face track to provide a single track level embedding, which is in turn used to perform non-temporal face clustering. Recent approaches, such as Huo & van Zyl (2020), adopt 3D convolutions inside the feature extractor to model temporal identity information better. Our method utilizes temporal information in a more flexible way, thus allowing the use of any feature encoder architecture.

**Temporal Track Constraints based Video Face Clustering:** A large body of methods focuses on generating identity labels through the creation of positive image pairs. They track a given face across consecutive frames and negative pairs through co-occurring tracks. Approaches such as Cherniavsky et al. (2010); Kapoor et al. (2009); Yan et al. (2006) apply such temporal constraints in semi/fully supervised settings, whereas methods like Wu et al. (2013a;b); Tapaswi et al. (2014); Xiao et al. (2014); Dahake et al. (2021); Tapaswi et al. (2019); Somandepalli & Narayanan (2019); Aggarwal et al. (2022); Datta et al. (2018) use temporal constraints in an unsupervised manner, with the majority of them adopting some contrastive pair loss formulation. We significantly improve on this major trend by skipping negative pairs selection and thus any complex mining strategy for obtaining them.

**Deep Metric Learning:** Deep face clustering inherently relies on having embeddings of the same identity closer to each other and of different identities farther away in the representation space. Approaches such as Song et al. (2017); Cinbis et al. (2011); Zhang et al. (2016a); Yang et al. (2016); Law et al. (2017); Mensink et al. (2012) focus on optimizing such a space and require defining a face similarity metric to improve video face clustering performance. Most approaches incorporate a contrastive-based similarity metric, such as triplet loss, to help obtain an embedding space optimized for a given set of faces in a video. We adopt the metric defined in Eq. (2) that does not rely on any negative pairing, thereby avoiding any sub-optimality induced through incorrect negative pair selection. In this paper, we also utilize this metric for final video track clustering, which provides enhanced performance since the embeddings are optimized w.r.t. the metric itself.

**Joint Representation and Clustering:** Sharma et al. (2019) propose two methods, TSiam and SSiam, to generate positive and negative pairs for model finetuning on a set of given video faces. TSiam adopts track-level constraints, i.e., sampling faces within the same and co-occurring tracks for positive and negative pairing, respectively. SSiam mines hard contrastive pairs using a pseudo-relevance feedback (pseudo-RF) inspired mechanism Yan et al. (2006). Both methods employ com-

plex modules that depend on finetuned parameters to mine negative pairs. In contrast, our proposed method does not depend on negative pairs at all, making it much simpler and generalized. Also, while Sharma et al. (2019) incorporates a baseline final clustering algorithm, which depends on the number of clusters as initialization, we propose a novel final clustering algorithm that is directly linked to the model finetuning approach and requires no initial guidance. Zhang et al. (2016b) incorporate a Markov Random Field (MRF) model to assign coarse track cluster labels, used as weak supervision for iteratively training a feature encoder. Negative pairs are mined through specific temporal constraints to boot start optimization of MRF. We introduce a much simpler weak matching mechanism that removes any dependency on negative pairs, thus requiring no complex training as in MRF. Besides, our clustering algorithm requires no input number of known characters for determining final cluster labels.

## 3 METHOD

### 3.1 OVERVIEW

A high-level overview of our proposed method is shown in Fig. 2. The following subsections describe in detail the prominent components of our approach.

### 3.2 FACE TRACK PRE-PROCESSING

In a standard frame rate video, frame content within the same scene gradually varies w.r.t. its temporal neighboring frames. To exploit this temporal stability for face clustering, we first locally cluster detected faces in a video by motion tracking, akin to all major prior works Cinbis et al. (2011); Sharma et al. (2019); Wu et al. (2013b). This pre-processing stage consists of four components.

First, scene cuts are detected in the given video, which divides it into contiguous separate sections, here coined as shots. Each scene cut represents a major change in scene composition, either involving a camera angle or scene setting change. We employ a threshold-based scene cut detection algorithm implemented in PySceneDetect library API (2022). Second, we utilize a face detection algorithm to find all visible faces in each frame of the processed shots. We adopt RetinaFace Deng et al. (2020) as our face detector due to its current state-of-the-art benchmark performance.

Third, the detected face crops are evaluated for their face ID quality by thresholding facial attributes based on blurriness and crop size. Crops failing the quality test are directly labeled as *Unknown*. Fourth, detected faces within a given shot are locally linked into a face track using a motion tracking algorithm. We adopt the state-of-the-art method, BoT-SORT Aharon et al. (2022), to generate tracks. For each face track $t$, a face crop $I$ is sampled every 12-th frame, i.e., $t = [I_{t_1}, I_{t_2}, ...., I_{t_n}]$, where $t_n = 12 * n + f_1$ and $f_1$ denotes the original frame index for the first frame in the track's sampled set $t$. This particular frame interval assumes a video frame rate of 24 fps and ensures, in most cases, that there is a significant change in either facial pose and/or expressions through the track duration.

### 3.3 TASK OBJECTIVE FORMULATION

Following Gong et al. (2019); Wu et al. (2013a), we consider a set of all detected tracks within a given video, which can be denoted by $T = \{t_j | j = 1, 2, ...., N\}$ for a set of N tracks. The face clustering objective can be defined as assigning an unique cluster id $d$ for each track $t_j$, where all tracks with the same id belong to an unique facial identity in the set $T$. Note that the ground truth number of clusters is undefined. More formally,

$$t_j^d = \{-1, 1, 2, ...\}, \ \forall j = \{1, 2, ..., N\}, \tag{1}$$

where $d = -1$ indicates the *Unknown* face cluster. This cluster represents tracks with the majority of their faces flagged as failures in any of the previously mentioned face attributes tests or if the face ID model was uncertain about it. The latter case is detailed in Section 3.6.

### 3.4 SELF SUPERVISED MODEL FINETUNING

To adapt a large-scale pre-trained face ID model to a specific set of faces, we incorporate the notion of finetuning the model for that face track set. Traditional supervised finetuning would require human supervision, i.e. ground truth labels, which can be tedious depending on the number of tracks involved. To alleviate this problem, several approaches in the domain of self-supervised

feature learning (SSL) have recently been proposed van den Oord et al. (2018); Tian et al. (2020); He et al. (2020); Chen et al. (2020). Especially interesting are methods that only use positive pairs for contrastive-based learning Grill et al. (2020); Zhou et al. (2022b). Inspired by Zhou et al. (2022a), we adopt a self-distillation-based SSL method that uses a teacher-student mechanism and positive pairs.

First, we modify the technique to perform finetuning rather than training from scratch. As shown in Fig. 3, given the pre-trained face-ID model, which has no specific architecture limitations, we attach a randomly initialized multilayer perceptron (MLP) as a model head. For a Transformer model architecture Dosovitskiy et al. (2021), separate heads are attached for the class and patch token embeddings, respectively. The base model with the attached head(s) is duplicated to create a teacher branch, with the original one designated as the student branch.

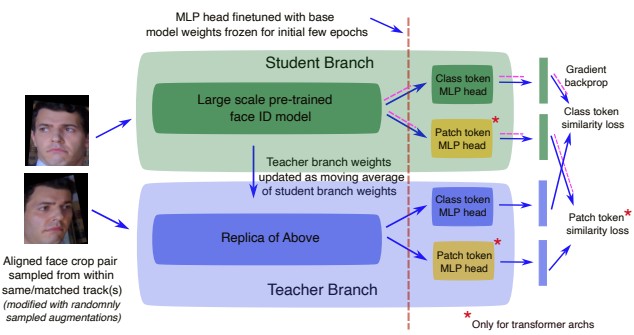

Figure 3: **Self-Supervised Model Finetuning:** Face crop pair sampled from within same/matched track(s) is passed through a student and teacher branch, respectively. Gradients w.r.t. similarity loss are backpropagated only through the student branch, while the teacher weights are updated as moving average of student weights. Random augmentations set includes horizontal flipping, rotation and color temperature variations.

Second, image pairs are curated from each track's sampled crop set. Images in a given pair are first passed through a set of augmentations, then an embedding pair is obtained as the output of the two constructed branches. We adopt a similarity loss to compare these embeddings, presented as follows:

$$L_{ssl} = -1 * softmax\left(\frac{embed_t - c}{temp}\right) * \log(softmax(embed_s)), \quad (2)$$

where $embed_t$ and $embed_s$ are the embeddings from teacher and student branches, respectively. Here, $c$ denotes a rolling average teacher embedding computed across training batches, and $temp$ is a fixed softening temperature. Respective loss gradients are backpropagated through the student branch weights only, while the teacher branch weights are updated via a moving average of the student weights at regular training intervals. Please refer Appendix A.3 for additional implementation details.

As the branch heads are randomly initialized, each of the branch's base model is frozen for an initial training phase. First, the heads are updated separately, akin to the description above. Then, both the base model and the heads are updated. Such a structured training regime encourages the model branch heads to produce robust and consistent embeddings for a given facial identity across the observed range of poses, expressions, lighting and appearance changes, thus improving overall clustering performance for that specific video.

### 3.5 COARSE TRACK MATCHING

Since a given face track is limited to being within a shot, there is no significant variation in lighting and/or face appearance across the track, which theoretically puts an upper bound on the model learning capacity. However, for real-world scenarios such as those likely to be observed in movies and TV series, such parameters can vary greatly throughout the video. To account for such variations and facilitate further model learning, we perform fully automated coarse matching of tracks across the entire dataset. Image pairs generated from such coarse-matched tracks enable the model to better adapt to specific lighting and appearance variations encountered in a given face across the entire dataset. This notion is supported by our experimental findings in Section 4 and Appendix H.

For coarse track matching, we leverage multiple sampled crops of the same identity in a given face track. To model this track crop distribution, we found empirically that fitting a multivariate normal distribution on all track crop embeddings works the best. Mathematically, we adopt

$$N_{t_j}(\mu_{t_j}, \Sigma_{t_j}) = (2\pi)^{-d/2} * \det(\Sigma_{t_j})^{-1/2} * \exp\left(\frac{-1}{2}(x - \mu_{t_j})^\top \Sigma_{t_j}^{-1}(x - \mu_{t_j})\right), \quad (3)$$

where $\mu_{t_j} \in \mathbb{R}^d$ and $\Sigma_{t_j} \in \mathbb{R}^{d \times d}$ are the mean and covariance matrix for the $j^{th}$ face track, computed using all its sampled face crop embeddings. Here, $d$ denotes the number of dimensions of the fitted distribution, which equals the dimension of track crop embeddings.

To automatically set a custom threshold value for matching a given face track to other neighboring tracks in the model learned embedding space, we resort to the probability density function (pdf) values of a given track's crop embeddings. Specifically, the pdf values of all crop embeddings are computed w.r.t. their parent track's fitted distribution using Eq. (3). We then consider the lowest 25% of these values and compute their mean. This provides a customized matching threshold, which is illustrated as a ring around the track's distribution in Fig. 4. To get coarse matches for a given track, we compute a mean embedding for every other dataset track and its corresponding pdf value w.r.t. the given track's fitted distribution. If a neighboring track's pdf value is equal to or

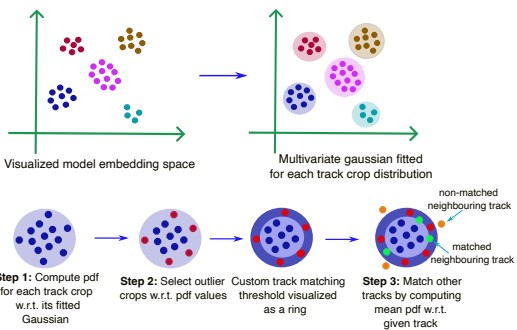

Figure 4: **Coarse Face Track Matching:** A Multivariate Gaussian is fitted to every track crop distribution. Then, a custom track matching threshold is computed using outlier crops pdf values. Neighboring tracks having a mean pdf value higher than custom threshold are soft-matched with the given track.

higher than that given track's custom match threshold, then the track pair's distributions have significant overlap, hinting at a strong face identity match.

Further, given coarse matches for every dataset face track, we curate face crop pairs across these track matches for the next iteration of model finetuning. In particular, for each face crop in a given track, we randomly sample a track from a set of its coarsely matched tracks. The image pair is created by randomly sampling a crop from within the sampled track. We empirically found that this image pairing mechanism works better than other more complex strategies such as thresholding inter track euclidean/cosine distances. If a track has no coarse matches, then we create pairs from within the same track.

### 3.6 TRACK FACE QUALITY ESTIMATION

In complex face identification scenarios, excluding bad quality crops/tracks becomes essential for coarse track matching and final clustering. Bad face quality of a given track often relates to model uncertainty, which can result in false track matches during the coarse matching phase or wrong clustering in the final stage. To automatically estimate the face quality of track crops, we adopt SER-FIQ Terhörst et al. (2020), which utilizes a dropout layer to determine consistency in embeddings predicted by the model for multiple iterations of the same face crop input. For bad quality crops, the learned model that is uncertain about them would predict embeddings with high variance, thus resulting in a low-quality score.

To compute the face quality score for a given track (tqs), we adapt SER-FIQ to work on a track level by obtaining scores for each track crop and averaging them. To detect bad quality tracks, we adopt the median absolute deviation (MAD) Leys et al. (2013) to detect roughly the lower 5% outlier tracks based on dataset track score distribution and compute a threshold value. Please refer to Appendix B for details. Low quality score tracks are filtered out from coarse track matching and final clustering modules and label their final track cluster IDs as *Unknown*.

### 3.7 FINAL CLUSTERING

To cluster tracks across common identities, we utilize the SSL loss function in Eq. (2) as an embedding similarity metric. Prior works incorporate Euclidean or other pre-defined distance metrics to compare model embeddings Sharma et al. (2019); Tapaswi et al. (2019). Our proposed metric has significant benefit. A finetuned model's embedding space is directly optimized w.r.t. this metric, thus making it optimal for evaluating embedding similarity. As such, there are no implicit assumptions made about the space through generic distance functions. Unlike other methods that define a global matching threshold Defays (1977), our approach automatically computes a custom threshold per face track, which measures how well the model has learned about that particular facial identity.

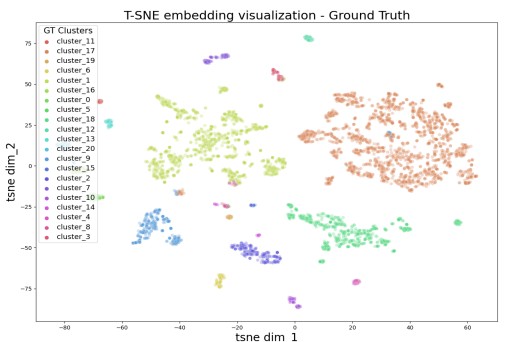 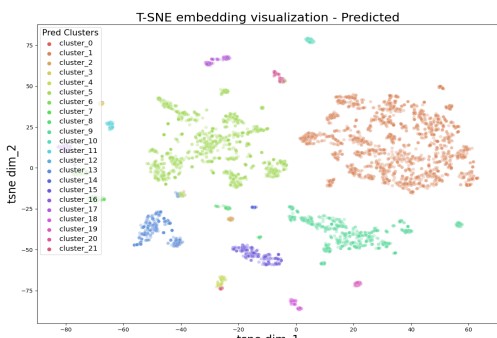

Figure 5: Comparative t-SNE embedding visualizations der Maaten & Hinton (2008) on **MovieFaceCluster:The Hidden Soldier** dataset. *Left:* Ground truth, *Right:* Our method. Each dot in the diagram above represents the finetuned model's extracted embedding for a given face crop $I_{t_n}$ in a given track's sampled crop set $t$. Face embeddings assigned to a given color constitute a single cluster. Our method predicts almost perfectly the cluster designations (22 clusters) w.r.t. ground truth (21 clusters).

The proposed algorithm is described in Algorithm 1. We begin by creating all possible pair combinations among sampled face crops within a given track. Here, we exclude tracks filtered out by the track face quality estimation module described in Section 3.6. A pair's similarity value is computed via the loss metric by passing the respective face crops through each of the model branches. As the loss metric is not commutative, a mean value is computed by alternatively sending both images through each of the branches. A custom track matching threshold is set as the average of all curated pair similarity values in the given track. This threshold represents quantitatively how well the model matches face crops belonging to a given common facial identity since the crops are part of the same track. We repeat this step for all tracks in the dataset.

The next stage comprises of merging tracks in a bottom-up approach, akin to Hierarchical Agglomerative Clustering (HAC) Defays (1977). Initially, each track is assigned an individual cluster, and tracks are iteratively merged if they satisfy a matching criterion. It involves creating all pairwise combinations of face crops across both tracks for a given candidate pair of tracks. Given the similarity (loss) value for every crop pair combination, taking a mean across them provides a matching potential value for that candidate track pair. If this match potential is lower than either of the track's custom match thresholds, then the track pair is considered a positive match. This process is repeated for all possible track combinations in the dataset. All positively matched pairs are searched for common tracks so that they can be combined together into a bigger cluster. For example, if track pair 1,2 and 2,3 are matched, then tracks 1, 2, and 3 are combined together. This entire matching process is run in an iterative fashion. For later iterations, where clusters could have more than one track, mean track embeddings are considered instead of a combined set of face crops for cluster pair matching, to avoid exponentially growing match computations. The algorithm terminates when no new clusters are merged in the new iteration.

## 4 RESULTS

We present our experimental analysis on popular benchmark datasets and our curated movie dataset.

---

**Algorithm 1:** Face Track Clustering

**Input:**
**Filtered Face Tracks** $T = \{t_j \mid j = 1, 2, ..., N\}$
$\ni t_j = \{I_{t_1}, I_{t_2}, ...I_{t_n} \mid t_n = 12 * n + f_1\}$,
**finetuned model** $\theta_{ft}$, **Similarity metric** $S$

**Stage 3.1 (Compute track crop embedding set):**
**for** $t_j$ *in* $T$ **do**
    **for** $I_{t_n}$ *in* $t_j$ **do**
        |   $E_{t_n} \longleftarrow \theta_{ft}(I_{t_n})$
    **end**
    $t_{jE} = \{E_{t_1}, E_{t_2}, ..., E_{t_n}\}$
**end**
$T_E = \{t_{1E}, t_{2E}, ...t_{NE}\}$

**Stage 3.2 (Compute custom track threshold):**
**for** $t_{jE}$ *in* $T_E$ **do**
    $Sim_{t_{jE}} = \{S(E_{t_l}, E_{t_m}) \;\; \forall \; (l, m) \in {}^nC_2\}$
    $Thres_{t_j} \longleftarrow \; mean(Sim_{t_{jE}})$
**end**
$T_{thres} = \{Thres_{t_1}, Thres_{t_2}, ..., Thres_{t_N}\}$

**Stage 3.3 (Perform track clustering):**
**Initialize**
  $i = 0, C = \{C_j = \{t_{jE}\} \; \forall \; j = 1, 2, ..., N\}$
**repeat**
    **for** $C_j$ *in* $C$ **do**
        **for** $C_k$ *in* $C$ *if* $k \neq j$ **do**
            $Sim_{jk} \longleftarrow$
                $mean(\{S(t_{jE_a}, t_{kE_b})$
                $\forall \; (a, b) \in {}^nC_2\})$
            **if** $Sim_{jk} < Thres_{t_j}/ \; Thres_{t_k}$
            **then**
                |   $C_j \longleftarrow \; merge(C_j, C_k)$
            **end**
        **end**
    **end**
    $C = link\ merges\ (C)$
    $N_{C_i} = ClusterCount(C)$
    Repeat Stage 2 for new merged cluster set $C$
    $i = i + 1$
**until** $N_{C_i} - N_{C_{i-1}} = 0$;

**Output: Clustered track IDs** $C$

---

### 4.1 BENCHMARK DATASETS

Following prior work Bäuml et al. (2013); Tapaswi et al. (2019); Sharma et al. (2019), we evaluate our proposed method on TV series episodes of Big Bang Theory (BBT) and Buffy The Vampire Slayer (BVS), specifically the first six episodes of BBT season 1 and BVS season 5, respectively. BBT is a TV series with primarily indoor setting, a cast of 5∼8 different characters and 625 average face tracks per episode. Here, all shots include wide-angle scenes, and faces are relatively small. The most common face ID challenges are pose and lighting variations. BVS poses different challenges. The main cast comprises 12∼18 characters, and there are 919 average face tracks per episode. Shots are mainly captured outdoors, and scenes are dark. It also has more close-up shots and, thus, larger face sizes. Detailed statistics on these datasets are shown in Tab. 1 in Bäuml et al. (2013). To compare against previous methods, we use the same face detection, tracking, and clustering labels as provided in Tapaswi et al. (2019); Sharma et al. (2019). Tab. 1 compares our method with state-of-the-art methods on BBT and BVS, respectively.

### 4.2 METRICS

We define two primary metrics at face track level for evaluating video face clustering performance, namely Weighted Cluster Purity/Accuracy (WCP) and Predicted Cluster Ratio (PCR). WCP is defined as the fraction of common identity tracks in a predicted cluster, weighted by the cluster track count. PCR is the ratio between the predicted cluster and ground truth cluster count. Note that a ratio closer to 1 is deemed better. Here "Unknown" is considered as a separate ground truth cluster if available.

| Method | BBT Episode | | BVS Episode | |
|---|---|---|---|---|
| | S1E1 | Combined | S5E2 | Combined |
| Wu et al. (2013a) | 66.48 | - | - | - |
| MLR Bäuml et al. (2013) | 95.18 | 83.71 | 61.27 | 66.37 |
| HMRF Wu et al. (2013b) | - | - | 50.30 | - |
| WBSLRR Xiao et al. (2014) | - | - | 62.70 | - |
| Zhang et al. (2016b) | - | - | 92.10 | - |
| TSiam Sharma et al. (2019) | 96.40 | - | 92.46 | - |
| SSiam Sharma et al. (2019) | 96.20 | - | 90.87 | - |
| BCL Tapaswi et al. (2019) | 98.63 | 89.63 | 79.76 | 83.62 |
| CP-SSC Somandepalli & Narayanan (2019) | - | - | 65.20 | - |
| MvCorr Somandepalli et al. (2021) | - | - | 97.70 | - |
| **Ours*** | **99.72** | **97.35** | **99.10** | **96.10** |

Table 1: WCP/Clustering Accuracy on BBT Season 1 and BVS Season 5. The results on all six episodes combined are presented in column 2 and 4 for BBT and BVS respectively. *Note that here, we use ArcFace-R100 Deng et al. (2019) as our pre-trained base model.

### 4.3 MOVIE DATASET

Mainstream movies present challenges for face clustering due to extreme pose, illumination, and appearance variations. Considering the lack of significant benchmark datasets in the academic research community, we present a new movie benchmark dataset named **MovieFaceCluster**, containing a collection of movies, hand-selected by film post-production specialists, with unique face clustering challenges. Refer to Appendix E for additional details on the dataset. Given the set of movies, we run the preprocessing mentioned in Section 3.2 to obtain a specific track dataset for each movie. We hand-label each track with an ID using the main character cast from that track's parent movie. Also, false detections and extreme unidentifiable tracks are pre-filtered out using the track face quality estimation module mentioned in Section 3.6. Tab. 2 provides detailed statistics on this curated dataset.

Tab. 2 also compares our method to other similar state-of-the-art approaches on this benchmark to provide empirical evidence of our algorithm's effectiveness, specifically for extremely challenging cases. We achieve the best cluster accuracy (WCP) and predicted cluster ratios closest to 1 for all individual movies. Fig. 5 additionally shows a t-SNE plot visualization of our learned embeddings for one of our dataset movie and Fig. 1 illustrates some hard case clustered tracks using our proposed method on MovieFaceCluster dataset.

| Statistics | Movie | | | | | | | | | Total |
|---|---|---|---|---|---|---|---|---|---|---|
| | An Elephant's Journey (2019) | Armed Response | Angel Of The Skies | Death Do Us Part (2019) | American Fright Fest | The Fortress | Under The Shadow | The Hidden Soldier | S.M.A.R.T. Chase | |
| Track Count | 562 | 119 | 319 | 395 | 457 | 917 | 143 | 594 | 113 | 3619 |
| Character Count | 18 | 14 | 29 | 7 | 37 | 64 | 9 | 21 | 10 | 209 |
| Avg. Track Length (secs) | 3.625 | 2.912 | 4.064 | 4.775 | 3.702 | 5.357 | 5.458 | 4.058 | 5.190 | 4.349 |
| Method | Weighted Cluster Accuracy (%) & Pred Cluster Ratio (Pred / GT) | | | | | | | | | |
| TSiam Sharma et al. (2019) | 90.7 & 1.44 | 84.9 & 1.36 | 77.1 & 0.62 | 92.9 & 1.57 | 89.3 & 0.83 | 68.6 & 0.69 | 71.8 & 2.11 | 90.7 & 1.33 | 79.6 & 1.70 | - |
| SSiam Sharma et al. (2019) | 88.1 & 1.61 | 86.6 & 1.21 | 75.5 & 0.59 | 94.4 & 1.28 | 86.2 & 0.78 | 71.1 & 0.73 | 68.3 & 2.33 | 88.7 & 1.24 | 82.3 & 1.80 | - |
| Zhang et al. (2016b) | 91.4 & 1.33 | 85.2 & 1.50 | 73.4 & 0.62 | 90.8 & 0.71 | 91.5 & 0.86 | 65.3 & 0.77 | 73.1 & 2.00 | 92.6 & 1.19 | 85.8 & 1.70 | - |
| Ours | **97.2 & 1.11** | **94.1 & 0.93** | **85.9 & 0.72** | **98.0 & 1.14** | **97.6 & 0.92** | **89.3 & 1.02** | **82.5 & 1.88** | **98.5 & 1.04** | **93.8 & 1.50** | - |

Table 2: *Top:* Statistics on our **MovieFaceCluster** Benchmark Dataset. *Bottom:* Quantitative comparisons on each individual movie. All computed results are based on incorporating ArcFace-R100 Deng et al. (2019) as the pre-trained base model. We outperform SoTA methods w.r.t. cluster accuracy and predicted cluster ratio. For implementation details on comparative methods please refer to Appendix A.4.

## 5 ABLATIVE ANALYSIS

We ablate the central components of our method and analyze limitations and future directions.

**Model Finetuning** We ablate on the effectiveness of finetuning a generic face ID model to a given set of face tracks as part of our proposed method. Tab. 3 provides a comparison of clustering performance with and without using the model finetuning module. Note that our proposed final clustering algorithm depends on the similarity metric learned during the finetuning stage. As such, to compare both methods in a fair way, we adopt a common baseline clustering algorithm, i.e., HAC with average linkage and cosine distance metric. Performing model finetuning results in roughly 6% increase in cluster accuracy, which underlines its usefulness.

**Final Clustering Algorithm** We further ablate on the performance of our final clustering approach vs. baseline algorithm, i.e., HAC with average linkage and cosine distance metric in Tab. 4. Here, we keep the model finetuning stage constant in all methods to compare fairly. As for HAC, we take the mean of a given track's sampled crop embeddings to obtain a representative track embedding. We further ablate on the loss function as a similarity metric. Specifically, we compare our final clustering algorithm with using Euclidean and Cosine distances as similarity metrics. Our approach involving the loss metric outperforms all other methods.

**Generic face ID Model Architectures** In Tab. 5, we ablate on our approach's generalization capabilities to incorporate any generic face ID model, fairly agnostic to its architecture class. Specifically, we compare some prominent face ID models from both Convolutional Neural Network (CNN) and Transformer architecture classes incorporated as part of our method, against using them (w/o finetuning) along with baseline clustering method (HAC). Regardless of the incorporated face ID model, our finetuning method provides roughly a $5 \sim 12\%$ performance boost, which underlines our method's capability to adapt to and improve any generic face ID model.

| Method | Cluster Accuracy (%) |
|---|---|
| Baseline - (Non-Finetuned) | 86.10 |
| **Ours - (Finetuned)** | **91.52** |

Table 3: Ablation for model finetuning module, using ArcFace-R100 as the base model. Experiments are performed on **MovieFaceCluster: The Hidden Soldier** dataset and HAC as final clustering algorithm.

| Clustering Algorithm | Similarity Metric | Cluster Accuracy (%) | Cluster Ratio (Pred/GT) |
|---|---|---|---|
| Baseline (HAC) | Cosine | 91.52 | 1.43 (30 / 21) |
| Ours | Cosine | 93.70 | 2.0 (42 / 21) |
| Ours | Euclidean | 96.50 | 3.5 (74 / 21) |
| **Ours** | **Loss Func.** | **98.50** | **1.04 (22 / 21)** |

Table 4: Ablation for final clustering algorithm, compared with baseline HAC, and using pre-defined metrics within our algorithm. Experiments are performed on **MovieFaceCluster: The Hidden Soldier** dataset and using ArcFace-R100 as base model.

| Face ID Model | Cluster Acc. (%) | |
|---|---|---|
| | Baseline | Ours |
| FaRL-P16 Zheng et al. (2022) | 78.7 | **90.2** |
| VGGFace2-R50 Cao et al. (2018) | 84.2 | **95.7** |
| ArcFace-R100 Deng et al. (2019) | 86.1 | **98.5** |
| AdaFace-R100 Kim et al. (2022) | 86.9 | **98.4** |

Table 5: Ablation for various base face ID models incorporated in our method. We perform comparisons using our proposed method and pre-trained model + baseline clustering (HAC). Experiments are performed on **MovieFaceCluster: The Hidden Soldier** dataset.

**Limitations and Future Work** Usage of a generic face ID model means that any pre-existing model biases may also be propagated through our method. For example, if the generic model has learned an incorrect similarity between two distinct facial identities, then our algorithm might adapt to it and provide a false positive cluster for that given pair. A future direction could be to automatically detect such biases, such that a given pair's embeddings are specifically pulled apart. This could be done by incorporating an outlier detection technique based on pair similarity values for a cluster's tracks. Also, given that we finetune on a set of face tracks, it might not be optimal for real-time applications depending on the track set size. A future extension of our work could be to perform clustering on person rather than face bounding boxes, so that clustering is based on additional body cues rather than just focusing on the facial features. Comprehensive movie datasets such as MovieNet Huang et al. (2020) could be used to train large-scale pretrained models for the same.

## 6 CONCLUSION

We present a novel video face clustering algorithm that specifically adapts to a given set of face tracks through a fully self-supervised mechanism. This helps the model better understand and adapt to all observed variations for a given facial identity across the entire video without any human-in-the-loop label guidance. Our fully automated approach at video face clustering specifically helps avoid any sub-optimal solutions that maybe induced from non-intuitive user-defined parameters. In addition, using a model-learned similarity metric over generic distance functions helps provide state-of-the-art video face clustering performance over other competing methods. Extensive experiments and ablation studies on our presented comprehensive movie dataset and traditional benchmarks underline our method's effectiveness under extremely challenging real-world scenarios.

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

## A  IMPLEMENTATION DETAILS

### A.1  MODEL ARCHITECTURE

As part of our method, any CNN or Transformer architecture can be incorporated as the base Face ID model. For the attached MLP head, we incorporate a 3-layer network with GELU activation Hendrycks & Gimpel (2016). The length of the head's output feature embedding is kept the same as the base model's embedding length. Unnormalized model output embeddings are used for both training and the final clustering stage. Also, teacher branch head weights are initialized separately from their student counterpart. When using a transformer class base model, the head is shared between the class and path token embeddings in both student and teacher branches respectively (Fig. 3).

### A.2  DATA PRE-PROCESSING

Generation of tracks as part of given video dataset curation for finetuning and/or clustering purposes involves processing the video through the four stage pipeline mentioned in detail in Section 3.2. To further augment variations in a sampled face crop pair, 'global' and 'local' views are created from the original crops. Specifically, crops are randomly resized and cropped with a scale between 0.7 and 1.0 to create multiple global crops, while a scale between 0.4 and 0.6 provides local crop variants. Given a total of 6 views for each non-cropped original pair, we set the global and local crop count to 2 and 4 respectively. The loss for a given non-cropped original pair is computed using all possible pairs within each global and local crop group separately. We further apply horizontal flipping and color temperature variations as additional augmentations.

Face alignment is performed for each face crop prior to the model finetuning and final clustering stages, in order to further enhance model learning of facial features and in turn for better face clustering performance. For this, a given face crop is first resized to the pre-trained model's expected input image size. Then, five point face landmarks are predicted for that crop (we incorporate the landmarks provided as an auxiliary output by the face detection model RetinaFace Deng et al. (2020) as we empirically find them to be quite accurate). Given the facial landmarks, a similarity transform is computed w.r.t. a mean landmark template. An affine warp is then computed to align the face within the face crop.

### A.3  MODEL FINETUNING

For the initial finetuning iteration, we train the model for 30 epochs. Note that the first 10 epochs are dedicated to finetuning the branch heads. All subsequent iterations include 10 training epochs without individual head training stages since the heads aren't randomly reinitialized for every new iteration. The teacher branch weights are updated with the exponential moving average of the student branch weights after every epoch. We adopt AdamW optimizer Loshchilov & Hutter (2019) and an initial learning rate (lr) of $1 \times 10^{-4}$. We use cosine decay scheduling and reduce the lr to a final value of $1 \times 10^{-5}$. The initial lr is linearly warmed up for the first 5 epochs for each iteration from a starting value of $5 \times 10^{-6}$. Experiments were conducted using a Nvidia A10 GPU with 24 GB VRAM, running CUDA 12.0, with code implemented in PyTorch 1.13 Paszke et al. (2019). For finer details on the loss function and training hyperparameters, please refer to Zhou et al. (2022a).

### A.4  BASELINE METHODS IMPLEMENTATION DETAILS

**Implementation for TSiam and SSiam Sharma et al. (2019)**  The methods were implemented in Pytorch 1.13 Paszke et al. (2019). For a fair comparison with other methods, we utilize ArcFace-R100 Deng et al. (2019) instead of VGGFace2 Cao et al. (2018) as the base Face ID model. Since finer details on training hyperparameters are unavailable, we assume standard values for batch size (32 for TSiam) and training epochs (100). Besides, no image augmentations are added during training. We remark that we generate results on the MovieFaceCluster dataset tracks, excluding any bad face quality tracks that the original method struggles to cluster. A global threshold which was empirically deemed to be optimal for the entire MovieFaceCluster dataset was set as the cut-off threshold for HAC clustering module. For additional implementation details (which weren't modified in our implementation), please refer to Sharma et al. (2019).

**Implementation for Joint Face Representation and Adaptation Zhang et al. (2016b)**  The approach was implemented in Pytorch 1.13 Paszke et al. (2019). For a fair comparison with other methods, we adopt ArcFace-R100 Deng et al. (2019) instead of DeepID2+ Sun et al. (2015) as the base Face ID model. We implement a Markov Random Field (MRF) approach in Python from scratch for face clustering. For additional implementation details (which weren't modified in our implementation), please refer to Zhang et al. (2016b).

## B  ADDITIONAL DETAILS FOR TRACK QUALITY ESTIMATION

**Threshold value for filtering out bad face quality tracks**  For a given set of $N$ tracks, a quality score threshold is computed as follows:

$$thres(tqs(N)) = \overline{tqs(N)} - (2.7 \times MAD(tqs(N))) \tag{4}$$

where $tqs(t_n)$ is the face quality score computed for the $n^{th}$ track using technique detailed in Section 3.6, $tqs(N) = \{tqs(t_1), tqs(t_2), ..., tqs(t_j)\} \ \forall \ j = \{1, 2.., N\}$. Tracks having score lower than $thres(tqs(N))$ are filtered out from both coarse track matching and final clustering modules. The value of 2.7 was empirically found to work well for removing bad quality tracks from a wide range of movie track sets.

## C  ALGORITHM PSEUDO-CODE

We present the pseudo-code of the different steps in our algorithm in Algorithm 2. Mentioned stage numbers correspond to Fig. 2.

---

**Algorithm 2:** VideoClusterNet

---

**Input:**
**Face Tracks** $T = \{t_j | \ j = 1, 2, ..., N\}$
$\quad \ni t_j = \{I_{t_1}, I_{t_2}, ... I_{t_n} | \ f_{n+1} - f_n = 12\}$ (obtained from stage 1)
**pretrained model** $\theta_m$, **cluster iterations** $total \ iters$

**Stage 2: Self Supervised model finetuning**
$\theta_s, \theta_t \longleftarrow \ replicate(\theta_m)$
$\theta_s \longleftarrow \ \theta_s + attach \ head(\theta_h)$
$\theta_t \longleftarrow \ \theta_t + attach \ head(\theta_h)$
$T_{filtered} = T, \ T_{cm} = None$
**for** $i$ $in$ $total \ iters$ **do**
$\quad \theta_{s_i}, \theta_{t_i} \longleftarrow \ finetune \ model(\theta_{s_{i-1}}, \theta_{t_{i-1}}, T_{filtered}, T_{cm})$
$\quad T_{fq} \longleftarrow \ face \ quality \ estimation(T, \theta_{s_i})$
$\quad T_{filtered} \longleftarrow \ filter \ outliers(T, T_{fq})$
$\quad T_{cm} \longleftarrow \ track \ coarse \ matching(\theta_{s_i}, T_{filtered})$
$\quad \theta_{ft} \longleftarrow \ \theta_{s_i}$
**end**

**Stage 3: Fully Automated Face Track Clustering**
$T_{fq} \longleftarrow \ face \ quality \ estimation(T, \theta_{ft})$
$T_{filtered} \longleftarrow \ filter \ outliers(T, T_{fq})$
$C \longleftarrow \ cluster \ tracks(T_{filtered}, \theta_{ft})$

**Output: Clustered track IDs** $C$

---

## D  ADDITIONAL RESULTS FOR BBT AND BVS DATASETS

| Method | BBT Episode | | | | | | |
|---|---|---|---|---|---|---|---|
| | S1E1 | S1E2 | S1E3 | S1E4 | S1E5 | S1E6 | Combined |
| Wu et al. (2013a) | 66.48 | - | - | - | - | - | - |
| TSiam Sharma et al. (2019) | 96.4 | - | - | - | - | - | - |
| SSiam Sharma et al. (2019) | 96.2 | - | - | - | - | - | - |
| MLR Bäuml et al. (2013) | 95.18 | 94.16 | 77.81 | 79.35 | 79.93 | 75.85 | 83.71 |
| BCL Tapaswi et al. (2019) | 98.63 | 98.54 | 90.61 | 86.95 | 89.12 | 81.07 | 89.63 |
| Ours⋆ | **99.70** | **99.67** | **98.60** | **98.80** | **99.10** | **97.10** | **98.70** |
| Ours† | 97.40 | 97.10 | 93.80 | 94.10 | 95.60 | 90.80 | 92.45 |

Table 6: WCP/Clustering Accuracy on BBT Season 1. Here ⋆ / † denote not including / including filtered out tracks as an incorrect cluster, respectively. Note that we adopt ArcFace-R100 Deng et al. (2019) as our pre-trained base model.

| Method | BVS Episode | | | | | | |
|---|---|---|---|---|---|---|---|
| | S5E1 | S5E2 | S5E3 | S5E4 | S5E5 | S5E6 | Combined |
| HMRF Wu et al. (2013b) | - | 50.3 | - | - | - | - | - |
| WBSLRR Xiao et al. (2014) | - | 62.7 | - | - | - | - | - |
| TSiam Sharma et al. (2019) | - | 92.46 | - | - | - | - | - |
| SSiam Sharma et al. (2019) | - | 90.87 | - | - | - | - | - |
| CP-SSC Somandepalli & Narayanan (2019) | - | 65.2 | - | - | - | - | - |
| MvCorr Somandepalli et al. (2021) | - | 97.7 | - | - | - | - | - |
| MLR Bäuml et al. (2013) | 71.99 | 61.27 | 66.60 | 67.07 | 69.59 | 61.72 | 66.37 |
| BCL Tapaswi et al. (2019) | 92.08 | 79.76 | 84.00 | 84.97 | 89.05 | 80.58 | 83.62 |
| Ours⋆ | **96.30** | **99.10** | **98.70** | **97.43** | **99.00** | **96.78** | **98.10** |
| Ours† | 93.30 | 94.50 | 94.40 | 91.18 | 92.40 | 93.21 | 91.32 |

Table 7: WCP/Clustering Accuracy on BVS Season 5. Here ⋆ / † denote not including / including filtered out tracks as an incorrect cluster, respectively. Note that we use ArcFace-R100 Deng et al. (2019) as our pre-trained base model.

## E   MOVIEFACECLUSTER DATASET CURATION

The **MovieFaceCluster** dataset provides challenging face ID tracks within a set of hand-selected mainstream movies. These challenges involve large variations in pose, appearance, illumination and occlusions that are unavailable in any generic movie face ID datasets. To the best of our knowledge, this is the most comprehensive Video Face Identification dataset for movies to be open sourced. It consists of a total of 3619 face tracks across 209 different identities spanning nine movies. Each constituent movie has an unique set of characteristics in terms of number of characters, average track length, character age, ethnicity, and background environments, among other factors. Please refer to Tab. 8 for further specific details on it.

| Movie / TV Series | Specific Face ID challenges |
|---|---|
| An Elephant's Journey (2019) | Bright outdoor scenes, American cast |
| Armed Response | Low light scenes, facial occlusions with military helmets,sunglasses etc., African American and Middle Eastern Cast |
| Angel Of The Skies | Unique heavy occlusions with oxygen masks in bright settings |
| Death Do Us Part (2019) | Low light scenes, Extreme facial expressions like screaming, Extreme poses, Rapid movements, African American Cast |
| American Fright Fest | Facial occlusions like see through masks, sunglasses, extreme poses |
| The Fortress | Facial occlusions like headgear, Large main cast with primarily Asian characters |
| Under The Shadow | Low light scenes, Middle Eastern Cast |
| The Hidden Soldier | Low Light scenes, Asian Cast |
| S.M.A.R.T. Chase | Extreme lighting in some scenes, Asian Cast |
| Big Bang Theory (S1E01-06) | Mainly Indoor scenes in constant well lit environments, American cast |
| Buffy The Vampire Slayer (S5E01-06) | Overall darker scenes, American Cast |

Table 8: Specific Face ID challenges presented by each movie, as part of MovieFaceCluster dataset and literature benchmark datasets

Ground truth data is provided in form of tracks and face box spatial locations corresponding to a global movie frame index, w.r.t. a specific frame rate. These tracks are generated using the preprocessing module explained in detail in Section 3.2. As part of this processing stage, any bad quality tracks are discarded from the dataset. Also, each dataset movie consists of a mix of main and secondary characters. We remark that an unique face identity, which has at least two good quality tracks, is included as part of the movie dataset.

## F   VIDEO FACE CLUSTERING DATASET COMPARISON

In order to provide evidence to the uniqueness and advantages of our proposed MovieFaceCluster dataset compared to existing literature, in Tab. 9, we present a comparison across some dataset attributes that are critical for the face clustering task. In Fig. 6, we present a dataset percent histogram comparison across face crop quality scores computed per dataset track using a large scale pretrained Face ID model (ArcFace-R100 Deng et al. (2019)). Similarly in Fig. 7, we present a dataset percent histogram comparison for face crop parameters that are highly relevant for face clustering, namely scene lighting and face blur level. Scene lighting values are estimated as the average of lightness (L) parameter values in a given face crop image converted to HLS space. Face crop blur is estimated using a Singular Value Decomposition (SVD) based method Su et al. (2011). Significant differences weren't observed in face pose attribute across all datasets.

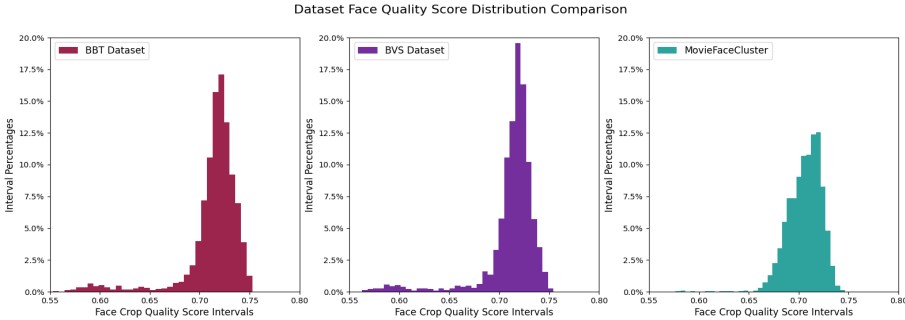

Figure 6: Comparison of track face crop quality score distributions across The Big Bang Theory (BBT), Buffy, The Vampire Slayer (BVS) and our MovieFaceCluster dataset. For a given dataset, face crop quality score is computed for each of its constituent track. It is estimated as the average of scores calculated for a given track's sampled crop set (using SER-FIQ Terhörst et al. (2020) and ArcFace-R100 Deng et al. (2019) as the pre-trained model for extracting embeddings). The distribution mean is relatively lower for our MovieFaceCluster dataset compared to other benchmark datasets, along with more bias towards lower quality score interval - 0.66 to 0.7. This provides empirical evidence towards our dataset containing more challenging cases for face clustering due to lower face quality scores.

| Dataset | TV series episode/Movie | Attribute | | | |
|---|---|---|---|---|---|
| | | Unique characters | Track Count | Unique ethnicity (other than White/Caucasian) (cast percentage) | Avg. Track Face Quality Score †† |
| The Big Bang Theory (BBT) | S01E01 | 6 | 647 | A (Minor) | 0.7193 |
| | S01E02 | 5 | 613 | A (Minor) | 0.7108 |
| | S01E03 | 7 | 562 | A (Minor) | 0.7094 |
| | S01E04 | 8 | 568 | A (Minor) | 0.7140 |
| | S01E05 | 6 | 463 | A (Minor) | 0.7177 |
| | S01E06 | 6 | 651 | A (Minor) | 0.7111 |
| | | Average: 6.33 | Total: 3504 | Unique Count: 1 (A) | Average: 0.7136 |
| Buffy The Vampire Slayer (BVS) | S05E01 | 12 | 786 | None | 0.7090 |
| | S05E02 | 13 | 866 | None | 0.7117 |
| | S05E03 | 14 | 1185 | None | 0.7150 |
| | S05E04 | 15 | 852 | None | 0.7125 |
| | S05E05 | 15 | 733 | None | 0.7081 |
| | S05E06 | 18 | 1055 | None | 0.7142 |
| | | Average: 14.5 | Total: **5477** | Unique Count: 0 | Average: 0.7120 |
| MovieFaceCluster | An Elephant's Journey | 18 | 562 | None | 0.7112 |
| | Armed Response | 14 | 119 | AA (Major), ME (Major) | 0.7085 |
| | Angel Of The Skies | 29 | 319 | None | 0.7150 |
| | Death Do Us Part (2019) | 7 | 395 | AA (Major) | 0.7177 |
| | American Fright Fest | 37 | 457 | AA (Minor) | 0.7098 |
| | The Fortress | 64 | 917 | A (Major) | 0.6918 |
| | Under The Shadow | 9 | 143 | ME (Major) | 0.7134 |
| | The Hidden Soldier | 21 | 594 | A (Major) | 0.7056 |
| | S.M.A.R.T. Chase | 10 | 113 | A (Major) | 0.7110 |
| | | Average: **23.22** | Total: 3619 † | Unique Count: **3** (A,AA,ME) | Average: **0.7062** |

Table 9: Specific dataset attribute comparisons across BBT, BVS and our MovieFaceCluster dataset. A: Asian, AA: African American, ME: Middle Eastern characters. † Only tacks that contained decent quality face crops were added as part of each movie dataset. Bad face crop quality and background character tracks were discarded. †† Score computed as average of all track crop quality scores as part of a given TV series episode/movie. A given track quality score is computed as average of quality scores for each of its sampled crops, using SER-FIQ Terhörst et al. (2020) and ArcFace-R100 Deng et al. (2019) as the pre-trained model for extracting embeddings. A lower average quality score would mean that the given dataset contains on average more challenging cases for face clustering

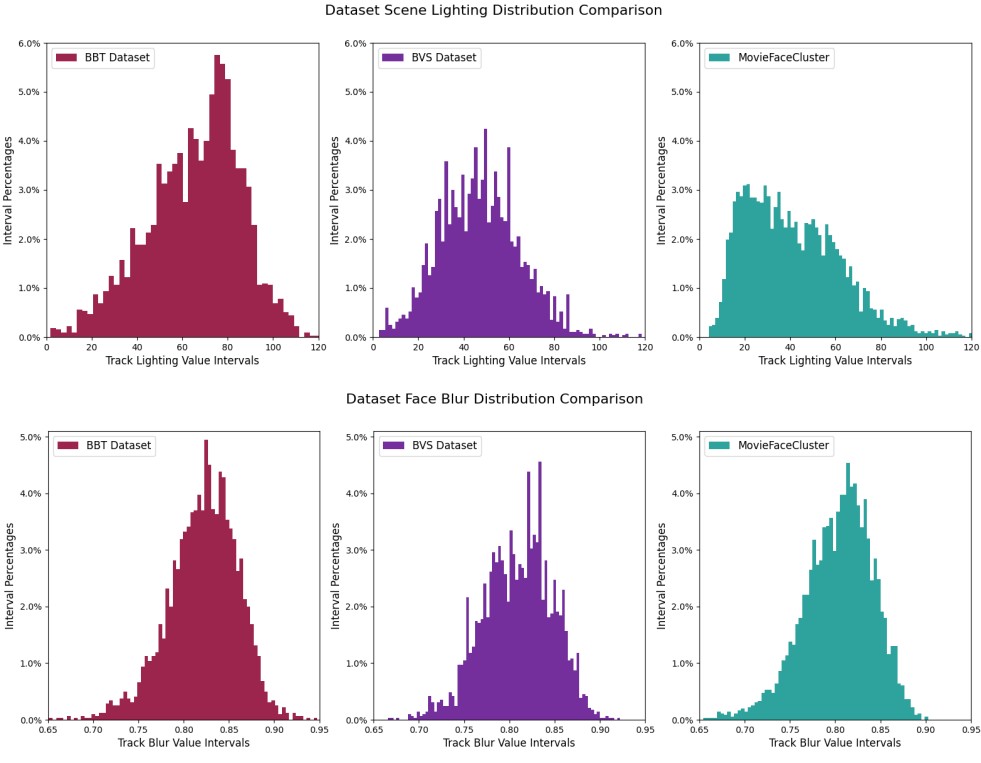

Figure 7: Dataset attribute histogram comparisons for scene lighting and face bluriness across The Big Bang Theory (BBT), Buffy, The Vampire slayer (BVS) and our MovieFaceCluster dataset. Each respective dataset attribute is computed per track, taken as average of each of its sampled crops attributes. For Scene Lighting, MovieFaceCluster has higher distribution variance compared to literature datasets, with more bias towards darker lighting (values have positive correlation to amount of track scene lighting). For Face Blur, MovieFaceCluster has higher sample count in 0.65 to 0.75 value range compared to literature datasets, providing evidence of having higher level of bluriness in track face crops (Blur values have negative correlation to amount of blur present in face crops).

With reference to Tab. 9, our proposed MovieFaceCluster dataset has on average lot more unique character faces and significantly better cast racial diversity. Also, it manages to obtain a much lower average face quality score, alluding to the fact that our dataset contains on average more challenging data samples w.r.t. face clustering/identification task. Fig. 6 further strengthens this argument by providing evidence of higher percent of dataset tracks having lower face quality scores within our dataset. In addition, Fig. 7 signifies, firstly that MovieFaceCluster has higher variance in scene lighting across its face samples with bias towards lot darker scenes which helps provide harder scenarios for face clustering. Secondly, for facial blur, our dataset has higher count of track samples with larger amount of facial blur compared to literature datasets, which again helps provide more challenging scenarios for face clustering.

## G TRAINING AND EVALUATION TIMINGS

Our proposed algorithm comprises two main stages from a computation standpoint: 1) Model SSL finetuning and 2) Final clustering. Tab. 10 presents run times of each of these stages for each movie of **MovieFaceCluster** dataset.

| Statistics | Movie | | | | | | | | | |
|---|---|---|---|---|---|---|---|---|---|---|
| | An Elephant's Journey(2019) | Armed Response | Angel Of The Skies | Death Do Us Part (2019) | American Fright Fest | The Fortress | Under The Shadow | The Hidden Soldier | S.M.A.R.T. Chase | Average (Per Track) |
| Track Count | 562 | 119 | 319 | 395 | 457 | 917 | 143 | 594 | 113 | - |
| Finetuning Iterations | 5 | 6 | 5 | 8 | 10 | 9 | 6 | 5 | 5 | - |
| Model Finetuning (mins) | 193.76 | 38.45 | 110.16 | 397.24 | 298.41 | 645.06 | 57.21 | 307.12 | 48.19 | 0.579 |
| Final Clustering (secs) | 66.36 | 6.74 | 28.42 | 40.82 | 61.94 | 167.32 | 13.93 | 81.32 | 9.13 | 0.132 |

Table 10: Training iteration count and runtimes for SSL finetuning and final clustering computed on MovieFaceCluster dataset. Here, the training iteration count represents the SSL finetuning iteration at which the process was terminated, and final clustering was performed.

## H SSL FINETUNING ITERATION ABLATION

This section provides ablation results for the Self-Supervised Learning (SSL) finetuning iteration parameter, performed on the MovieFaceCluster:The Hidden Soldier dataset. Our overall experiments showed that a minimum of 5 iterations are required to obtain optimal results. Additional few iterations (e.g., 2 or 3) are necessary for harder datasets. This is assuming the first iteration is run for about 30 epochs, with each succeeding one comprising 10 epochs. For iteration 10 and above, our method might over-cluster the dataset, i.e., it finds sub-clusters within the optimal clusters, optimizing for variants of a given character. Note that the first SSL

| SSL Finetuning Iteration | Accuracy / WCP (%) | Pred Cluster Ratio/ PCR (Pred / GT) |
|---|---|---|
| 1 | 52.56 | 0.476 (10/21) |
| 2 | 72.12 | 0.571 (12/21) |
| 3 | 91.38 | 1.114 (24/21) |
| 4 | 96.93 | 1.000 (21/21) |
| **5** | **99.50** | **1.048 (22/21)** |
| 6 | 99.41 | 1.095 (23/21) |
| 7 | 99.47 | 1.095 (23/21) |
| 8 | 99.50 | 1.190 (25/21) |
| 9 | 99.50 | 1.238 (26/21) |
| 10 | 99.41 | 1.238 (26/21) |

Table 11: Ablation for SSL finetuning iterations parameter, presented for **MovieFaceCluster:The Hidden Soldier** dataset. Finetuning until iteration 5 provides optimal results in terms of both metrics.

finetuning iteration involves image pair generation from within the same track only. For all succeeding iterations, image pairs are generated across coarse-matched tracks, facilitated through the coarse track matching process detailed in Section 3.5.

# I    ADDITIONAL T-SNE VISUALIZATIONS

Figs. 8 to 10 present additional t-SNE learned embedding visualizations for select movies that are part of the **MovieFaceCluster** dataset.

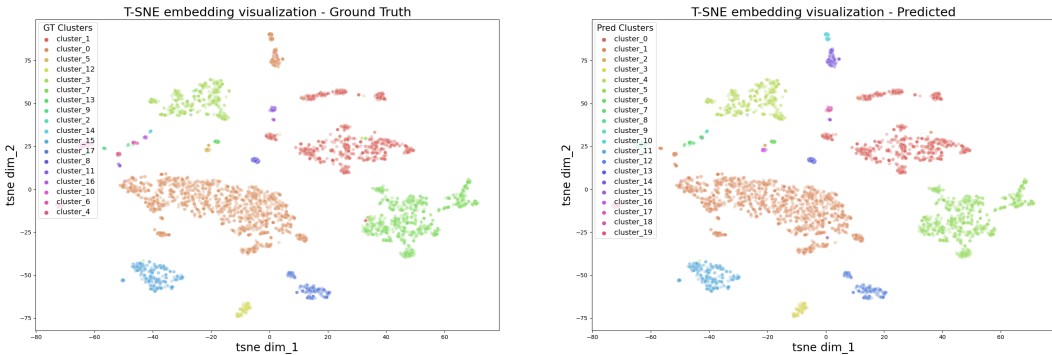

Figure 8: Comparative t-SNE embedding visualizations on **MovieFaceCluster:An Elephant's Journey** dataset. *Left:* Ground truth, *Right:* Our method. Each dot in the diagram above represents the finetuned model's extracted embedding for a face crop $I_{t_n}$ in a given track's sampled crop set $t$. Face embeddings assigned to a given color constitute a single cluster.

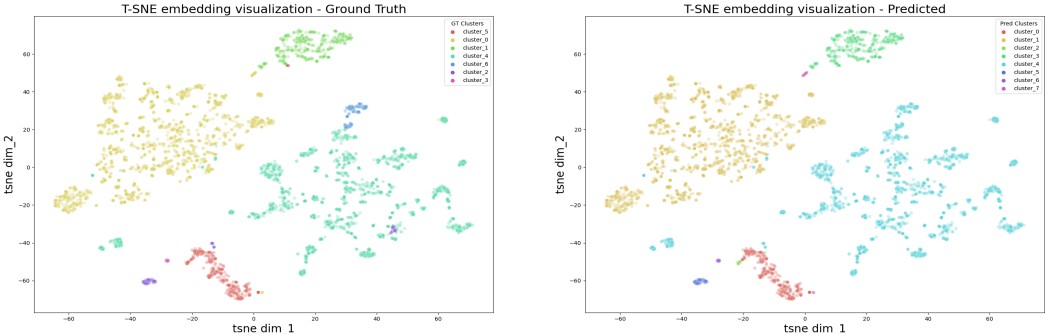

Figure 9: Comparative t-SNE embedding visualizations on **MovieFaceCluster:Death Do Us Part** dataset. *Left:* Ground truth, *Right:* Our method. Each dot in the diagram above represents the finetuned model's extracted embedding for a face crop $I_{t_n}$ in a given track's sampled crop set $t$. Face embeddings assigned to a given color constitute a single cluster.

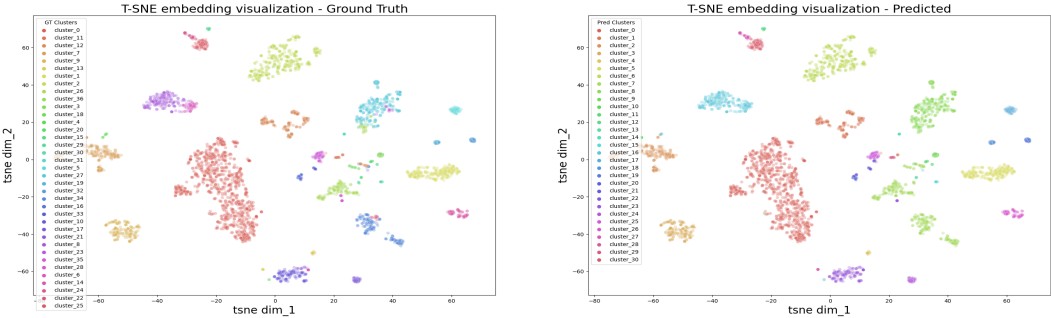

Figure 10: Comparative t-SNE embedding visualizations on **MovieFaceCluster:American Fright Fest** dataset. *Left:* Ground truth, *Right:* Our method. Each dot in the diagram above represents the finetuned model's extracted embedding for a face crop $I_{t_n}$ in a given track's sampled crop set $t$. Face embeddings assigned to a given color constitute a single cluster.

## J  ADDITIONAL HARD CASE CLUSTER VISUALIZATIONS

Fig. 11 presents an additional selection of hard case cluster results obtained with our proposed method on the **MovieFaceCluster** dataset.

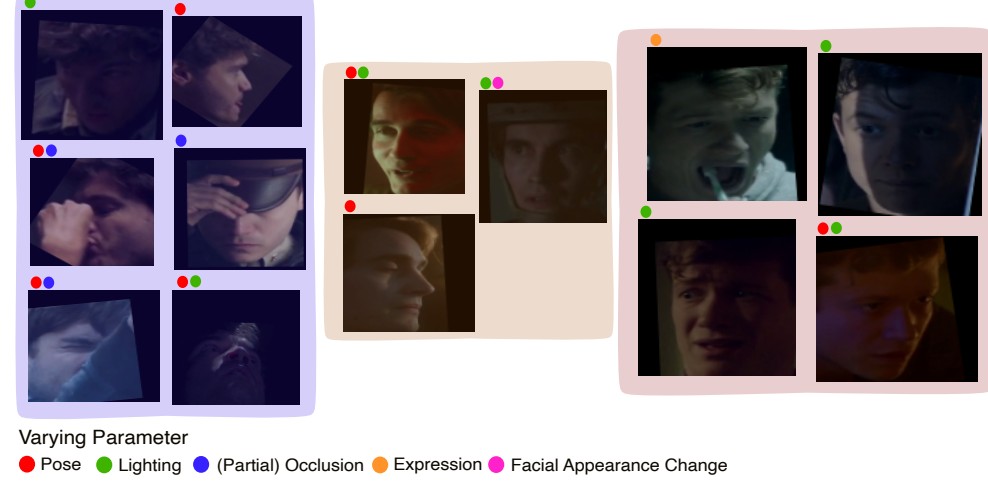

Figure 11: Select hard case clusters obtained through our proposed method on **MovieFaceCluster** dataset. The term "varying parameter" depicts the dominant image attributes that are particularly challenging for a given face crop. It isn't part of the available dataset annotations but simply mentioned for enhanced reader understanding.

