# OpenReview forum: "VideoClusterNet: Self-Supervised and Adaptive Face Clustering for Videos"
_ICLR.cc/2024/Conference — Submitted to ICLR 2024_

### Official Review · Reviewer_B82Y · 2023-10-31

**Soundness:** 2 fair
**Presentation:** 2 fair
**Contribution:** 2 fair
**Rating:** 5
**Confidence:** 4

**Summary:**

The authors propose a face clustering algorithm for videos that involves  a self supervised learning process to adapt a generic face ID model to the target video and also a clustering algorithm based on the adapted model. In addition, a new benchmark is proposed for the task of video face clustering. However, while the method of joint face representation adpation and clustering for video has been used in several prior works, such as in the related paper(Sharma et al. (2019), Zhang et al.(2016b)) mentioned by the authors, its novelty is not clearly established. On the other hand, the motivation for proposing the benchmark is not sufficiently explained.

**Strengths:**

1. Quite detailed experiments and good performance compared to current methods.
2. Automatic face clustering is important and practical for many video editing application but lacks sufficient attention in the research community. One of the reasons is the lack of good benchmarks. This work makes a contribution towards addressing this.

**Weaknesses:**

1. Compared to related works that use joint face ID model adaptation and clustering, the uniqueness of this work is not stated clearly. Is the performance increase comes from better face ID model adpatation or just the following clustering method?  If better face ID model, is it because of the teacher-student branch method used in this work or other perspectives?

2. Why we need a new benchmark? The existing ones not challenging enough for practical use? Why? Any quantitative/qualitative comparison among the porposed one and existing ones? The statement should be put clearly in the paper.

**Questions:**

Please refer to the section of weaknesses.

---

> ### Author Response · Authors · 2023-11-18
> **Response to Reviewer B82Y**
>
> We thank the reviewer for providing a thorough review of our paper. This would help us greatly in improving the quality and legibility of our manuscript. Below we address specific questions/concerns raised in the weakness section:
>
> 1. **Clarification on method novelty:** We kindly refer the reviewer to the “Joint Representation and Clustering” paragraph in Section 2 regarding the comparison of our proposed method with closely related methods. We believe that our method’s performance increase is a result of both the self-supervised (SSL) model finetuning and the final clustering modules. Empirical evidence for this fact can be observed in Section 5 (Ablation Analysis). Specifically, Table 3 and Table 4 prove the individual effectiveness of our model finetuning mechanism and clustering algorithm, respectively. Unlike closely related prior work, the model adaptation stage is more effective majorly due to the self-distillation procedure and its sole dependence on positive match pairs. Our final clustering algorithm directly benefits from the model finetuning stage by incorporating the SSL training objective as a distance metric. Such a learned metric helps boost the algorithm’s performance as it efficiently evaluates inter-track distances directly in an embedding space, which is optimized for reduction in the same identity distances. Also, any sub-optimal performance that may be induced due to any user-defined matching thresholds is avoided through the adaptive custom threshold computed for each track. Such a thresholding mechanism helps the clustering algorithm automatically adjust to how effectively the model can match a given track’s identity across a true positive match pair. We will update our paper with this detailed explanation for the reader’s enhanced understanding of our method’s salient novelty.
>
> 2. **Need for new benchmark dataset:** We kindly refer the reviewer to the second last paragraph in Section 1 (Introduction) where we detail a few primary reasons for the need for a comprehensive movie benchmark dataset for video face clustering. To justify it further, the problem we address in the paper, and its application for TV series/movies has seen comparatively lesser interest in the last few decades as opposed to other Computer Vision related tasks. Also, complicated usage and copyrights for TV series/movies meant that providing open source datasets based on them wasn’t feasible or otherwise costly and time-consuming w.r.t. providing annotations. As a result, the majority of related approaches in this domain end up evaluating their method on a very specific TV series/movie dataset while not being able to provide means to access it publicly. A direct consequence is thus not being able to have fair comparisons with prior works, forcing authors to evaluate across a large set of niche datasets. We present our dataset with this view in mind and sincerely hope that it will largely benefit the research community in this niche domain.
> Finally, we refer the reviewer to our response to the first reviewer (**RNxJ**)’s concern regarding dataset analysis and comparison. We also refer the reviewer to Appendix J, which summarizes the uniqueness and advantages of our proposed dataset over existing ones.
>
> We would appreciate it if you could let us know of any further questions/concerns that you may have and if any finer clarification is required for any of our responses listed above. If not, we would greatly appreciate it if the reviewer could consider increasing his/her overall rating.

---

> > ### Author Response · Authors · 2023-11-23
> > **Checkin For Any Further Concerns**
> >
> > Given that the discussion period ends today, we wanted to checkin and see if there are any more concerns or further clarifications required for any of our above mentioned points. If not, we would like to thank the reviewer for taking time to review our work and provide some valuable feedback. We sincerely hope to have answered all your concerns/questions! Thanks!

---

### Official Review · Reviewer_wXAX · 2023-10-31

**Soundness:** 2 fair
**Presentation:** 2 fair
**Contribution:** 2 fair
**Rating:** 5
**Confidence:** 4

**Summary:**

This paper presents a self-supervised algorithm for clustering face tracks in a video using a generic face ID model. A coarse track matching method is used to extract positive tracks for fine-tuning the face ID model. The fine-tuned model's embedding space is used to evaluate the similarity between face tracks, and an adaptive thresholding mechanism is used for the final clustering step. To evaluate the proposed model, a movie dataset is curated, and the results demonstrate its state-of-the-art performance.

**Strengths:**

- The annotation effort for the movie dataset is substantial.
- The method is novel in its elimination of the need for manual selection of a clustering threshold.
- The pre-processing step includes cutting-edge building blocks such as RetinaFace for face detection and SER-FIQ for face quality assessment.
- The noticeable efforts to replicate baselines in PyTorch are commendable.
- Limitations and future works are discussed.

**Weaknesses:**

Missing important details regarding fine-tuning: it is unclear how the data is divided into train/validation sets during the model fine-tuning stage. The self-distillation fine-tuning step proposed requires multiple tracks from different temporal steps to ensure adequate appearance variations. However, the number of tracks required in a video and how it impacts the model is not discussed.

The authors claim that the positive track pair construction step is independent of any ground truth labels or temporal motion track constraints. This statement is confusing as it contradicts the requirement of ground truth labels.

Unfair comparison with baselines due to advanced pre-processing: the pre-processing techniques used in this paper rely on advanced algorithms such as PySceneDetect (2022), RetinaFace (2020) and SER-FIQ (2020). In contrast, most of the baselines in Table 1 were established before 2020. There is no investigation into how these advanced pre-processing steps affect the clustering performance of the proposed method or the baselines. The observed performance improvement could be attributed to more accurate scene detection or face detection rather than the proposed method itself.

Data, training code and baseline re-implementation code are not promised to be open-sourced for reproducibility.

**Questions:**

See weaknesses.

**Details Of Ethics Concerns:**

The release of the proposed MovieFaceCluster dataset might breach the copyright of these movies.

---

> ### Author Response · Authors · 2023-11-18
> **Response to Reviewer wXAX [Part 1/2]**
>
> We thank the reviewer for taking the time to provide a thorough and detailed review of our paper. This would help greatly improve the quality and legibility of our manuscript. Below we address some specific questions/concerns posed by the reviewer in the weakness section:
>
> 1. **Missing details on model finetuning module:** We would first like to state that the proposed model finetuning module (details in Section 3.4) is fully self-supervised, i.e., completely independent of any manual ground truth labels. This facilitates the selection of all available tracks in a given video for the finetuning process. Also, the model finetuning hyperparameters such as training epochs, batch size, and learning rate (see Appendix A) are kept constant and are agnostic to the training dataset statistics. This removes dependence on a traditional train and validation set based mechanism for performance evaluation. These hyperparameters have been carefully tuned across a wide range of video datasets to ensure convergence of the fine-tuned model. This also helps the model learn all observed natural variations across a given set of dataset face crops. An alternate way to describe this finetuning mechanism is that in our self-supervised setup the training and validation/test set are indeed exactly the same, but where performance validation isn’t essential.
> Also, certain image augmentations (detailed in Appendix A.2) are applied during the finetuning process. This ensures that the model observes enhanced facial variations during its training and helps reduce its primary dependence on natural variations present in the dataset. In addition, since the model is finetuned on all face tracks that need to be clustered, it gives the model more flexibility to slightly overfit the training dataset if the track count is low. Both of these factors make the finetuning process fairly agnostic to the number of tracks and the available natural variations within a given video dataset. Table 2 shows empirical evidence of this fact wherein the track count ranges between 119 and 917 across our movie dataset. Our method manages to achieve significant performance improvements over benchmark methods across all the movies despite these track count variations. We hope that these insights help the reviewer better understand the salient novelty and merits of our proposed method. We will update future manuscript versions with this explanation accordingly.
>
> 2. **Positive track pair contribution clarification:** We thank the reviewer for mentioning the ambiguity in one of our listed paper contributions and apologize for the confusion. We would like to provide clarification about the second contribution in the introduction. Our method only relies on positive pairs in the model finetuning/adaptation stage, a unique feature from prior works. Previous methods leverage negative track pairs and incorporate either manual ground truth labels or temporal track constraints to obtain them. We have updated our second contribution accordingly to reflect this clarification.
>
> 3. **Clarification on baseline comparison with advanced pre-processing modules:** For the comparisons with baseline methods listed in Table 1, we used the exact same set of face bounding boxes and motion track information provided by the benchmarked methods for a fair comparison. We skipped the usage of our track pre-processing modules (detailed in Section 3.2) and directly processed the available data through the model finetuning and final clustering modules. Note that this data was made publicly available by (Tapaswi et al. 2019)[1] at this link: https://github.com/makarandtapaswi/BallClustering_ICCV2019. We believe this facilitates a fair evaluation of our self-supervised clustering algorithm w.r.t. baseline methods and eliminates any influence of advanced pre-processing modules, specifically shot and face detection. We kindly refer the reviewer to this fact mentioned in Section 4.1 of our paper. Similarly, for a fair comparison on our dataset in Table 2, we use our track pre-processing module to evaluate both the baselines and our method on the same set of face boxes.
>
> **Continued in Part 2/2**

---

> ### Author Response · Authors · 2023-11-18
> **Response to Reviewer wXAX [Part 2/2]**
>
> **Continued from Part 1/2**
>
> 4. **Dataset and code availability:** We would like to assure the reviewer that the proposed MovieFaceCluster dataset will be open sourced to the benefit of the wider research community and to encourage further research into this relatively niche domain. As for the concern about movie copyright issues with our dataset release, the reviewer should rest assured that we possess valid internal usage licenses for all the movies listed in our dataset. We will release the dataset in a format such that it doesn’t breach any copyright or license restrictions. In its current state, along with ground truth identity information, we will release the bounding box and motion track annotations referring to each frame in every movie. This would enable the dataset user to obtain the face crops given source media. We hope this effectively addresses the reviewer’s concern about the dataset's legal compliance. We will also work on making face crop data itself available in the next dataset version. We also confidently believe that we have provided thorough implementation details (in Appendix A) to accurately reproduce our proposed method. In addition, we will make available the re-implementation of the baseline code (accurate up to the best of our knowledge) upon the paper's publication.
>
> We would appreciate it if you could let us know of any further questions/concerns that you may have and if any finer clarification is required for any of our points listed above. If not, we would greatly appreciate it if the reviewer could consider increasing his/her overall rating.
>
> References:
> [1]: Makarand Tapaswi, Marc T. Law, and Sanja Fidler. Video face clustering with unknown number of clusters. In International Conference on Computer Vision (ICCV), pp. 5027–5036. IEEE, 2019.

---

> > ### Author Response · Authors · 2023-11-23
> > **Checkin For Any Further Concerns**
> >
> > Given that the discussion period ends today, we wanted to checkin and see if there are any more concerns or further clarifications required for any of our above mentioned points. If not, we would like to thank the reviewer for taking time to review our work and provide some valuable feedback. We sincerely hope to have answered all your concerns/questions! Thanks!

---

### Official Review · Reviewer_RNxJ · 2023-11-01

**Soundness:** 3 good
**Presentation:** 3 good
**Contribution:** 2 fair
**Rating:** 5
**Confidence:** 4

**Summary:**

This paper presents a self-supervised video face clustering approach that adapts to challenging variations in facial pose, lighting, and expressions by fine-tuning a generic face ID model to learn robust facial embeddings progressively. Furthermore, it introduces a parameter-free clustering algorithm that automatically clusters facial features based on the fine-tuned model embeddings without the need for user-defined thresholds or initial cluster numbers. In addition, a movie face clustering benchmark dataset MovieFaceCluster is provided to better evaluate the performance of video face clustering algorithms in real-world scenarios.

**Strengths:**

- The paper is well-motivated and easy to follow.
- The proposed method is reasonable and feasible.
- Compared to the prior work and baseline, the proposed method achieves competitive results.

**Weaknesses:**

- The paper does not provide enough information about the proposed dataset, particularly in terms of presenting its uniqueness and advantages. There is no visualizations or statistical analyses to demonstrate distinctions from existing datasets. Furthermore, the description of dataset annotations is unclear. It remains uncertain whether the term "Varying Parameter" mentioned in Figure 1 is associated with dataset annotations.
- The authors did not compare their dataset with existing movie person identification (PI) datasets, such as MovieNet [1], which includes annotations suitable for PI tasks.
- While the paper claims not to require pre-defined parameters, fixed values are still set in the quality assessment and coarse track matching modules to generate adaptive thresholds. However, these fixed values lack empirical or theoretical support.
- Regarding the t-SNE embedding visual comparison on the MovieFaceCluster dataset in Figure 5, the visualizations generated by the proposed method seem to closely approximate the ground truth, which exhibits some anomalous clusters. Authors may explore the limitations of proposed method and present failure cases so as to provide more in-depth insights into video face clustering.
- Most of the methods used for comparison are outdated, with only one introduced within the last three years.
- Please ensure the consistency of citation formats. For instance, there is an inconsistency in the citations of Table 1 in Section 4.1, where both 'Table 1' and 'Tab. 1' are used.

Ref:
[1] Huang, Qingqiu, et al. "MovieNet: A Holistic Dataset for Movie Understanding." European Conference on Computer Vision. 2020.

**Questions:**

See the above weaknesses.

---

> ### Author Response · Authors · 2023-11-18
> **Response to Reviewer RNxJ [Part 1/2]**
>
> We would first like to thank the reviewer for taking the time to provide a thorough and detailed review of our paper. This would greatly help us improve the quality and legibility of our manuscript. Below we provide some specific comments on questions/concerns pointed out in the weakness section:
>
> 1. **Dataset analysis and comparison:** In addition to the details on our proposed dataset presented in Section 4.3 and Appendix E, we have updated our paper to provide a comprehensive analysis and comparison of our dataset with existing literature in Appendix F. This includes a comparison of some key dataset attributes like unique character and track count, cast racial diversity, and average track face quality score (computed using our face quality estimation module presented in Section 3.6). We also present a detailed histogram comparison of the dataset face quality distribution to give more insights into each dataset’s average quality score. In addition, we provide dataset histogram comparisons across critical attributes, specifically scene lighting and facial blur. We are confident that this revision further strengthens our argument of providing a challenging, more diverse movie face clustering dataset. We sincerely thank the reviewer for his comments on this specific aspect of our paper. Also, in Figure 1 the term “varying parameter” denotes the dominant image attributes that are significant outliers for a given face crop. It isn’t part of the available dataset annotations and is used simply for enhanced reader understanding. We have added this clarification in the updated paper version in Appendix J.
>
> 2. **Comparison with MovieNet dataset:** Even though MovieNet dataset [1] comprises a wide variety of annotations on a large number of mainstream movies, we would like to kindly point out that its annotations aren’t directly useful for our task of video face clustering. It unfortunately consists of person bounding boxes (covering the entire character's body) and not face bounding boxes, which are essential for our task. Furthermore, complex post processing could be used to detect face bounding boxes in each dataset frame and match them against the ground truth person boxes, in order to associate the identity of each face. For this, the movie frame data would be essential to predict face boxes. To the best of our knowledge, neither the source movie dataset nor the links to the respective movies are available for download from https://movienet.github.io/. Due to these reasons, we strongly believe that a dataset comparison with MovieNet in its current state wouldn’t be fair for our intended task as it doesn’t contain relevant annotations for the same. We would however like to thank the reviewer for mentioning this specific dataset as it could potentially be relevant in a future extension of our work, wherein the video clustering is applied for person bounding boxes rather than just face boxes. We have updated our paper to acknowledge the same in the “Limitations and Future Works” paragraph in Section 5.
>
> 3. **Clarification for certain empirical system parameter values:** We would like to highlight here that the track quality estimation module (see Section 3.6) can be viewed as a pre-processing module for our final clustering algorithm (see Section 3.7). It is highly effective in filtering out non-identifiable face tracks that are often present in mainstream media content. Also, the coarse track matching module (see Section 3.5) is used in conjunction with the model finetuning module (see Section 3.4). We do acknowledge that both of the referenced modules contain minor empirically evaluated parameter values. However, we claim in our listed contributions (fourth contribution in the Introduction) that we specifically have a parameter-free video face clustering algorithm. Both of the aforementioned modules are essentially external to our core clustering algorithm. We would like to confidently reiterate that the core clustering algorithm is fully automated in the true sense. We would also like to provide clarification on specific values set for certain parameters in both referenced modules. For Equation 4 in the main paper, backed by empirical evidence, the value of 2.7 was first loosely set by fitting a Gaussian distribution onto the given set of track face quality scores. To select/filter the lower ~1.5% outliers, which are often less than the threshold of (mean - 2.8*std) in a Gaussian distribution, we sampled values in the range of 2.6 to 3.0 and empirically found that 2.7 worked optimally for our test set consisting of a wide range of movies.  **Continued in Part 2/2**

---

> ### Author Response · Authors · 2023-11-18
> **Response to Reviewer RNxJ [Part 2/2]**
>
> **Continued from Part 1/2**: Regarding the coarse matching module, the motivation to select crop pdf values in the lower 25% range was again based on empirical evidence. To effectively select the true outliers, which would result in a more accurate computation of a given track’s custom match threshold, we experimented with values ranging from 5% to 40%. We found out that setting it to 25% provides effective true positive track matches while avoiding any false positive matches (which is a critical performance parameter) over a wide range of movie track sets. We will update this in the future revised version accordingly.
>
> 4. **Anomalous clusters in t-sne visualization:** We believe that the majority of the anomalous clusters visible in Figure 5 for the ground truth visualization are a combination of two main factors. First, there exist certain sub-optimalities in the t-sne embedding method [2] used to find a two-dimensional visualization sub-space. That partly results in the appearance of those outlier clusters due to failure in finding the optimal subspace to have ground truth clusters in a more compact representation. Second, a few of the outlier clusters seen as part of the cluster sets: cluster_1, cluster_17, cluster_5 consist of tracks possessing higher than average difficulty for face clustering. As a result, these tracks appear distant from their respective cluster centers. We would like to highlight here that our method still accurately manages to assign such outlier clusters to their correct ground truth cluster even though they provide a harder face clustering challenge. We kindly refer the reviewer to the “Limitations and Future Works” paragraph in Section 5 of our paper, where we highlight in detail the core limitations of our work and specify future directions to improve on them.
>
> 5. **Literature method comparison:** There are two major selection criteria we used to choose the methods to benchmark our proposed method’s performance in Table 1. First, methods having results reported on The Big Bang Theory (BBT) and Buffy the Vampire Slayer (BVS) datasets were selected as we noticed a major overlap in the use of these two datasets for evaluation in the literature. Certain methods were omitted based on the fact that they only reported results on certain rare TV series datasets that weren’t publicly available for evaluation and/or didn’t have necessary usage rights. Secondly, methods that reported results on publicly available versions of BBT and BVS datasets were selected. For a fair comparison, all methods need to have a constant face track pre-processing module (details in Section 3.2) to ensure that clustering performance is evaluated on the same set of face tracks. Methods that did not satisfy these criterias also, unfortunately, didn’t provide publicly available implementations of their work, which prevented us from re-evaluating these methods on the available input data. For the same reason, we decided to re-implement certain closely related works ourselves for a fair comparison in Table 2. We remark that to the best of our knowledge, Aggarwal et al. (2022)[3] is the only paper published within the last 3 years that we haven’t reported in Table 1. It unfortunately failed to meet the second aforementioned criterion. We sincerely hope that this clarification helps the reviewer provide more insights into our evaluation methods selection criteria shown in Table 1. We would be open to any paper suggestions from the reviewer that may be closely related to our work that we might have missed.
>
> 6. **Inconsistency in citations:** We would like to thank the reviewer for mentioning this minor issue. We have resolved it in the updated paper.
>
> We would appreciate it if you could let us know of any further questions/concerns that you may have and if any finer clarification is required for any of the points listed above. If not, we would greatly appreciate it if the reviewer could consider increasing his/her overall rating.
>
> References:
> [1]: Huang, Qingqiu, et al. "MovieNet: A Holistic Dataset for Movie Understanding." European Conference on Computer Vision. 2020.
> [2]: Laurens Van der Maaten and Geoffrey Hinton. Visualizing data using t-sne. Journal of Machine Learning Research, 9(11), 2008.
> [3]: Abhinav Aggarwal, Yash Pandya, Lokesh A. Ravindranathan, Laxmi S. Ahire, Manivel Sethu, and Kaustav Nandy. Robust actor recognition in entertainment multimedia at scale. ACM International Conference on Multimedia (ACM MM), pp. 2079–2087, 2022

---

> > ### Comment · Reviewer_RNxJ · 2023-11-22
> > **Comment**
> >
> > Thank you for the clarification. My concerns regarding Q2, Q3, and Q4 were adequately addressed. Some concerns remain:
> >
> > 1. Dataset analysis and comparison (A1). The author states in Appendix E: "The MovieFaceCluster dataset provides challenging face ID tracks within a set of hand-selected mainstream movies. These challenges involve large variations in pose, appearance, illumination, and occlusions." Therefore, it would be better to focus visualizations or statistical analyses of the dataset on the variations in key attributes within face ID tracks, rather than solely analyzing faces.
> >
> > 2. Clarification on comparative methods (A5). I agree with the clarification regarding the comparison of methods on the MovieFaceCluster dataset (Table 2). However, there exist concerns regarding the selection criteria for the experimental dataset chosen for the proposed method. Given that the motivation behind the proposed method is to address challenging environmental variations, it is expected to demonstrate superior performance on more other datasets. Consequently, to better demonstrate the effectiveness of the proposed method, the authors should not restrict the dataset selection to BBT and BVS datasets.

---

> > > ### Author Response · Authors · 2023-11-22
> > > **Further Clarification On Specific Concerns**
> > >
> > > Thank you for your valuable feedback. We would like to provide further clarification on the remaining two concerns:
> > >
> > > 1. **Dataset analysis and comparison:** We would like to clarify that the dataset attribute comparisons presented in Appendix F are indeed based on Face ID tracks and not just individual faces. The caption of Table 9 and Figure 6 as well as the first paragraph of Appendix F provide a detailed description of how each attribute is calculated per track. Note that the set of face ID tracks (for each dataset) that have been used for performance evaluations in Tables 1 and 2 are the same as those used for dataset attribute comparisons in Figures 6 and 7. We will clarify this further in the main paper. In view of the above, we believe that our presented dataset analysis is very relevant and insightful since it highlights the uniqueness of our proposed dataset and the fact that it contains more challenging face ID scenarios.
> > >
> > >
> > > 2. **Benchmark Dataset selection:** We would first like to refer the reviewer to our response to reviewer **B82Y**, regarding reasons for proposing a new benchmark dataset. To highlight, we curate MovieFaceCluster dataset to enable method evaluations on challenging, in-the-wild face ID scenarios, which is lacking within current existing datasets. Also, it isn’t always feasible to evaluate our approach on some of the niche datasets since they aren’t open-sourced due to various reasons. Our proposed track pre-processing module (detailed in Section 3.2) could be used to obtain face ID tracks for each of these datasets. However, it wouldn’t result in a fair comparison due to possible differences in track pre-processing employed by the respective prior works.
> > > Our main focus is on comparing the baselines with our approach using a challenging dataset. In that regard, our proposed method achieves comprehensive performance improvement (on average 8% across all listed movies, Table 2) on a dataset that was specifically selected by film industry professionals to contain extreme face ID challenges (empirical evidence for the same is provided in Appendix F). Also, we remark that few of the prominent prior works, specifically Tapaswi et al. (2019) [1] and Sharma et al. (2019) [2] have provided their method comparisons only across BBT and BVS datasets. Baseline comparisons across BBT (easy), BVS (medium) and MovieFaceCluster (hard) datasets underline our proposed method’s state-of-the-art performance on varying complexities of face ID challenges.
> > >
> > > We hope to have provided further insights into our dataset selection and analysis across the different datasets. Please do let us know if all your concerns have not yet been fully clarified. And we thank you again for your prompt response.
> > >
> > > References:
> > > 1] Makarand Tapaswi, Marc T. Law, and Sanja Fidler. Video face clustering with unknown number of clusters. In International Conference on Computer Vision (ICCV), pp. 5027–5036. IEEE, 2019.
> > >
> > > 2] Vivek Sharma, Makarand Tapaswi, M. Saquib Sarfraz, and Rainer Stiefelhagen. Self-supervised learning of face representations for video face clustering. In International Conference on Automatic Face & Gesture Recognition, pp. 1–8. IEEE, 2019.

---

### Author Response · Authors · 2023-11-18
**General Comment To All Reviewers**

We deeply thank all the reviewers for their detailed reviews and comments to improve our manuscript. Overall we greatly appreciate all the constructive feedback provided by the reviewers.
Given the general concern about missing detailed comparisons of our proposed dataset with existing ones and its advantages and uniqueness, we have updated our paper with the same in Appendix F which provides clear empirical evidence for our dataset’s strengths and distinguishing attributes.
While searching for existing benchmark datasets for our method evaluation, we found out that while our method consistently performed well on a few existing TV series datasets, these datasets failed to provide in-the-wild challenges to effectively highlight our method’s strengths. We remark that quite a few far older baseline methods managed to perform equally as well on these datasets as some of the newer ones, which raises a case of performance saturation on these datasets. Thus, along with our novel state-of-the-art video face clustering method, we wanted to curate and present a more challenging dataset having in-the-wild scenarios to encourage the research community to improve on current stagnant baselines and advance the state of the art. Note that this dataset also helps showcase our method’s uniqueness and enhanced performance specifically for very challenging scenarios in mainstream movies, such as extreme facial pose, scene lighting, blur, and appearance changes. We truly hope that the reviewers would weigh equally these significant contributions of our work, both in terms of a novel state-of-the-art video face clustering algorithm and a far more challenging benchmark dataset for the same, thus enabling in-the-wild performance evaluations.

**Summary Of Updates In Revised Paper:**.

1] Update in **Introduction** section, second contribution (w.r.t. response to reviewer **wXAX** point 2).
2] Update to **Limitations and Future Works** paragraph in section 5 (w.r.t. response to reviewer **RNxJ** point 2).
3] Update to **Table 8** in Appendix E (w.r.t. response to reviewer **RNxJ** point 1 and **B82Y** point 2).
4] Addition of **Appendix F** (w.r.t. response to reviewer **RNxJ** point 1 and **B82Y** point 2).
5] Update to **Appendix J** (w.r.t. response to reviewer **RNxJ** point 1).

---

### Meta-Review · Area_Chair_Xwte · 2023-12-05

**Metareview:**

This paper was reviewed by three experts and received 5, 5, 5 as the ratings. The reviewers acknowledged that the proposed method is reasonable, and the experimental results are encouraging. However, the reviewers raised concerns about the comparison baselines being outdated and the lack of comparative analysis of the proposed dataset. Concerns were also raised regarding unfair comparison of the proposed method with the baselines due to the usage of advanced pre-processing techniques for the proposed method.

We appreciate the authors' efforts in meticulously responding to each reviewer’s comments, which helped clarify some of their concerns. We also appreciate the authors' efforts in updating the Introduction and Future Work sections of the paper, and adding more details and visualizations about the new dataset in the Appendix sections. However, during the AC-reviewer discussion period, it was mentioned that the paper will need a more thorough dataset analysis and more sufficient experimental evaluation, before it can be accepted for publication. In light of the above discussions, we conclude that the paper may not be ready for an ICLR publication in its current form. While the paper clearly has merit, the decision is not to recommend acceptance. The authors are encouraged to consider the reviewers' comments when revising the paper for submission elsewhere.

**Justification For Why Not Higher Score:**

The paper lacks a thorough experimental analysis with recent relevant baselines, which does not merit its acceptance at ICLR, considering its high standards.

**Justification For Why Not Lower Score:**

N/A.

---

### Decision · Program_Chairs · 2024-01-16

Reject